# Contribution of Asphalt Rubber Mixtures to Sustainable Pavements by Reducing Pavement Thickness

**DOI:** 10.3390/ma15238592

**Published:** 2022-12-02

**Authors:** Liseane Padilha Thives, Jorge C. Pais, Paulo A. A. Pereira, Carlos A. O. F. Palha, Glicério Trichês

**Affiliations:** 1Department of Civil Engineering, Federal University of Santa Catarina, Florianópolis 88037-000, SC, Brazil; 2Department of Civil Engineering, ISISE, University of Minho, 4800-058 Guimarães, Portugal; 3Department of Civil Engineering, CTAC, University of Minho, 4800-058 Guimarães, Portugal

**Keywords:** asphalt rubber, asphalt mixtures, cracking, performance, sustainable

## Abstract

Asphalt rubber mixtures have been used as pavement surface layers due to their ability to prevent early degradation, and are considered a sustainable option. This study performed analysis comparing the fatigue resistance of asphalt rubber mixtures of different combinations of asphalt bases, crumb rubber, and gradation, in order to estimate the performance of asphalt rubber mixtures as pavement surface layers. The methodology was developed in a laboratory and involved asphalt rubber production by continuous and terminal blend systems with different crumb rubbers and asphalt base types. Asphalt rubber mixtures with varying gradations and an unmodified asphalt mixture as a reference were produced. The mechanical behavior as a dynamic modulus and with respect to fatigue resistance was evaluated using a four-point bending test. In order to verify each of the asphalt rubber mixtures’ contribution as a surface layer, pavement structures were designed and their lifespans were compared. The findings showed that all asphalt rubber mixtures presented higher fatigue resistance than the reference. For pavement design, in comparison with the reference mixture, the thickness of the surface layer could be reduced by at least 50% while achieving the same life, proving its successful performance. This study demonstrated the effective contribution of crumb rubber from scrap tires as an asphalt modifier for producing sustainable mixtures with adequate fatigue performance.

## 1. Introduction

Structural distress in road pavements, mainly characterized by fatigue cracking and permanent deformation, has been a primary concern of road agencies, and the application of conventional asphalt mixtures cannot bypass the occurrence of distress under heavy traffic conditions. Improvements in road pavements have been achieved by using polymers during asphalt modification processes, extending pavement life, especially under conditions of heavy traffic and high temperature, demonstrating that the typical behaviors of asphalt can be enhanced by adding polymers.

In general, when blended with asphalts, polymers modify the asphalt material, improve its resistance at medium and high temperatures, and enhance its viscosity in intervals that allow workability at the plant and in the field [1]. Meanwhile, the performances of mixtures with modified asphalt also depend on the characteristics of aggregates, mixture gradation, and volumetric parameters [2].

The problems caused by the disposal of scrap tires are international in scope, and remain an environmental concern that many countries must handle [3]. From 1.6 billion new tires produced yearly, 1 billion scrap tires are generated, and it is estimated that only 100 million are forwarded to recycling facilities [4].

Scrap tires can be used in many applications, such as for fuel in cement industries and ground for sub-product processing. However, despite available technologies and efforts to deal with scrap tires by reuse and recycling, a significant quantity are disposed of in landfills [5].

Efficient public policies must be established to minimize the disposal of scrap tires in landfills or illegal dumps. In the United States, public policies to fund the recycling and management of scrap tires have been implemented. The policies contributed to tire recyclers’ investment in equipment and tire recycling facilities, resulting in increased production of crumb rubber [4]. In 2016, Canada established the Resource Recovery and Circular Economy Act to keep scrap tires out of landfills. Under this legislation, producers ensure that scrap tires are collected, recycled, or reused [6].

In Portugal, a similar policy was adopted, in which the producer of new tires is responsible for their collection, transport, and proper destination. Liability only ceases after an authorized entity attests to the scrap tire’s adequate destination [7]. In addition, Spain has the same obligations for tire producers regarding the destination of scrap tires, aiming to protect the environment and promote public awareness of waste prevention [8].

One alternative process to reduce improper scrap tire disposal is to crush the tires to obtain crumb rubber, which when added to asphalt by wet processes promotes rheological modifications and is called asphalt rubber [9,10]. Another option is to introduce crushed rubber into the asphalt mixture by a dry processes. A recent study conducted with asphalt mixtures incorporating crushed rubber via a dry process showed that the mixtures were less susceptible to aging than the reference. However, the authors asserted that meeting Swiss standard requirements for the production of these mixtures was challenging [11].

Crumb rubber from scrap tires can be obtained from ambient and cryogenic grinding. For use in asphalt modifications, it must fit a standard gradation [12]. The mechanical process (ambient grinding) is performed at an ambient temperature, resulting in crumb rubber with an irregularly shaped and elevated surface area [13]. In cryogenic grinding, scrap tires are frozen by liquid nitrogen to become brittle, and broken by a hammer mill. The crumb rubber resulting from this process presents smooth aspect particles with relatively small surface areas [14]. In addition, it is considered a cleaner and quicker process, providing operational crumb rubber with more fine gradations than the ambient method [15].

Asphalt rubber is obtained from wet processes, including continuous and terminal blend systems. In some countries, asphalt rubber refers by terminology to modified asphalt produced from a continuous blend system following the ASTM D6114 standard [16] allowing lower crumb rubber content (up to 10%), while terminal blending uses 15% to 20%.

Extended storage time during the process is an advantage of the terminal blend system. Generally, the asphalt rubber must be applied up to six hours after production in the continuous blend [16,17]. In both systems, the crumb rubber and conventional asphalt are mixed at a suitable controlled temperature for an appropriate time to promote changes for obtaining modified materials [18].

Fatigue cracking due to repeated loading has been recognized as a critical distress problem in flexible pavements at intermediate service temperatures. It is an essential element to be considered in flexible pavement designs and is mainly influenced by the asphalt mixture’s properties, especially its fatigue resistance [19,20].

Some authors have asserted that the prediction of fatigue life must consider the asphalt mixture characteristics such as aggregates, gradation curves, voids, and asphalt content. The environmental effects must also be measured and controlled [21,22]. The factors relative to the mixture characterize the material’s initiation, propagation, and failure phases [22]. The asphalt mixture’s mechanical behavior and stiffness (modulus) data are necessary to evaluate the pavement’s stress–strain response under traffic loading [23].

Asphalt rubber mixtures have been used successfully in various applications, and mechanical resistance improvements concerning permanent deformation, fatigue, and reflective and thermal cracking are described in the literature [24,25,26,27,28,29].

Under the conditions, asphalt rubber mixtures can reduce early pavement degradation, in addition to being considered environmentally friendly. From the point of view of sustainability, reuse or recycling of waste is always beneficial. Bueno et al. [30] complemented this view by expressing that despite the high performance of modified asphalts, the environmental disadvantages must be accounted for in the evaluation.

Wang et al. [31] affirmed that crumb rubber used as an asphalt modifier represents an effective method for recycling scrap tires and can contribute to sustainable management of solid waste. In addition, research in China proved that asphalt rubber mixtures reduced energy consumption by 47.18% and CO_2_ emissions by 17.06% compared with other mixtures produced with styrene–butadiene–styrene (SBS).

Yang et al. [32] evaluated emissions during production of mixtures in an asphalt plant at different temperatures. One mixture used conventional asphalt, and two were produced with asphalt rubber (12% crumb rubber). The asphalt rubber mixture was the traditional mixture; in another sample, an additive was included to reduce the temperature requirements. For all mixtures, the asphalt base was PG 58-28, and the mixing temperatures were 158 °C (conventional), 160 °C (traditional), and 133 °C (warm). The authors concluded that the total emissions strongly depended on the mixing temperature, and elevated temperatures resulted in increased emissions. The emissions from asphalt rubber mixtures were higher than conventional ones, particularly for xylene and toluene.

Zanetti and Farina [33] affirmed that asphalt mixtures generate emissions with polycyclic aromatic hydrocarbons and volatile organic compounds due to high temperatures during production. Their study evaluated the environmental performance of asphalt mixtures and toxicological effects on workers, via a life-cycle assessment (LCA). Air samples were collected during construction of a trial section in Italy. Two asphalt rubber mixtures with different gradations (gap and dense-graded) and a conventional mixture were compared. The LCA analyses demonstrated that asphalt rubber mixtures provided benefits in terms of energy saving, environmental impact, human health (human toxicity noncancerous effects excepted), the preservation of ecosystems, and the minimization of resource depletion. The also findings showed that the gap-graded rubber mixture (higher asphalt content) had a potential carcinogenic effect on workers that was 3.5 times higher than dense-graded mixtures (lower asphalt content) and 2.9 times higher than the conventional mixture. In assessment of the toxicological effects, the higher asphalt content resulted in a 1.3 times higher risk. Under these conditions, advantages can be attained, providing that the pavement is correctly designed with the possibility of reducing the surface layer’s thickness.

Milad et al. [34] stated that incorporation of crumb rubber in asphalt could contribute to sustainability, and previous studies proved that it enhances the mixture’s mechanical behavior. In opposition, authors have considered that such mixtures do not solve every pavement problem, and their performance depends on regional (temperature and climate) conditions. Some limitations were described: high service temperatures (mixture and compaction), more rigorous quality control, elevated costs, challenges related to reclaimed asphalt pavement (RAP) proportions, pollutant concentrations that include heavy metals, and toxic chemicals that affect workers’ health and the environment.

This study aims to perform a comparative analysis of the fatigue resistance of asphalt rubber mixtures relative to the asphalt base, crumb rubber, and gradation in order to estimate the performance of different mixtures as pavement surface layers. Moreover, we provide an method to reduce the disposal of scrap tires by their use in the production of asphalt rubber mixtures, and consequently concur with road-paving sustainability.

## 2. Materials and Methods

### 2.1. Method

The experimental program comprised production of asphalt rubber mixtures and their mechanical testing (modulus and fatigue). The operation program comprised five phases detailed in the method flowchart (Figure 1).

In Phase 1, tests were performed to characterize aggregates, crumb rubber, and base asphalt. For aggregates, gradation, particle shape, abrasion, water absorption, specific gravity tests, and cleaning tests were carried out. Ambient and cryogenic crumb rubbers were evaluated from gradation tests and image analyses using scanning electron microscopy (SEM). Base asphalts (PEN 50/70 [35] from Brazil and PEN 30/45 [36] from Portugal) were characterized using conventional tests (penetration, softening point, resilience, apparent viscosity, and aging by the rolling thin film oven test—RTFOT).

In Phase 2, asphalt rubber was produced by a continuous blend system in the laboratory (17% rubber content; processing time of 90 min at 180 °C). These asphalts were confectioned with two bases (PEN 50/10 [35] and PEN 30/45 [36]). In the mixing process, the base asphalt was first heated at 180 °C and then the crumb rubber was incorporated in portions for less than two minutes. The asphalt and crumb rubber were maintained under agitation (mechanized agitator with propeller), and the propeller speed was 250 to 300 rpm. After 90 min, the asphalt rubber was ready to use. Terminal blended samples had 15% and 20% rubber content, and were blended at a refinery where the base asphalt used was PEN 50/70. All asphalts were characterized in this phase to verify compliance with the specifications. Table 1 presents the designation of asphalt rubber according to the crumb rubber type, content, and production system.

Eleven asphalt rubber mixtures with three different aggregate gradations and a reference were produced in Phase 3 (Table 2). The reference mixture was produced using PEN 50/70 [35], designed according to the Brazilian specification DNIT grade “C” [37], the most commonly used in Brazil.

The gap-graded asphalt rubber mixtures followed the California Department of Transportation [38] and the Arizona Department of Transportation [39] standards, while the dense-graded mixture was designed to the Asphalt Institute mix type IV standard [40]. All mixtures were designed by the Marshall method.

After design, the asphalt mixtures were produced, and samples were obtained to perform mechanical tests (Phase 4). Dynamic modulus and fatigue resistance were evaluated using four-point bending tests. Master curves were constructed to describe the complex modulus of the asphalt mixtures. The influences on the mixture’s fatigue performance of the crumb rubber (ambient or cryogenic), base asphalt, asphalt rubber production system, and aggregate gradation were calculated.

The contribution to sustainable asphalt pavements was assessed by evaluating the necessary thickness of the surface layer in the pavement’s design, according to the performance of the mixture obtained in the mechanical tests. In this manner, it was possible to determine whether the pavement thickness could be reduced by choosing the appropriate asphalt rubber mixture (Phase 5).

### 2.2. Materials

#### 2.2.1. Base Asphalts Characterization

The base asphalts used to produce asphalt rubbers were classified according to penetration grade. The characterization test results are summarized in Table 3, showing that the asphalts fit the specifications and produced suitable asphalt rubber mixtures.

#### 2.2.2. Crumb Rubbers

Ambient and cryogenic crumb rubbers (Figure 2) wer compatible with the Arizona Department of Transportation’s gradation standard [46]. It can be observed that ambient rubber had particle sizes ranging from 0.05 mm to 1.2 mm, those in the cryogenic rubber ranged from 0.2 mm to 0.8 mm, and the surface areas were 19.3 m^2^/kg and 13.6 m^2^/kg, respectively.

Scanning electron microscopy (SEM) images (magnified 50 times) are presented in Figure 3. The ambient rubber particles (Figure 3a) present a spongy aspect, and an irregular structures of different sizes with formation of agglomerates (smaller particles adhering amongst themselves). In contrast, the cryogenically prepared sample (Figure 3b) presents smooth, cracked surfaces with angular corners. The aspects of the crumb rubber were similar to observations in research studies conducted by other authors [12,13].

#### 2.2.3. Aggregates

The aggregates (granite) and mineral filler to produce asphalt mixtures were from the north of Portugal and were classified as grade 1 (particle size from 6 to 12 mm), grade 2 (particle size from 4 to 10 mm), and grade 3 (particle size lower than 4 mm). The characterization tests results are shown in Table 4 (coarse aggregates) and Table 5 (fine aggregates), confirming that these aggregates are suitable for producing asphalt mixtures.

#### 2.2.4. Asphalts Rubber

Terminal blend asphalt rubbers were produced at a Brazilian refinery, using two respective rubber contents, i.e., 15% (A5015TB) and 20% (A5020TB), and 50/70 pen asphalt. Table 6 shows the characterization test results of terminal blend asphalt rubbers, and Table 7 shows the obtained results for continuous blend asphalt rubbers.

The tests and the standard limit specifications were the same for both asphalt rubber systems. The continuous blend asphalt rubbers were produced at the laboratory, using 17% crumb rubber content. The arrangement of two crumb rubber types and two base asphalts resulted in four modified asphalts. All asphalt rubbers meet the specification requirements [16] and can be used to produce asphalt mixtures.

#### 2.2.5. Asphalt Mixtures

Figure 4 shows the mixtures’ gradation curves, according to three specifications for asphalt rubber mixtures and one for a reference mixture.

After design, the asphalt mixtures were produced, and their volumetric parameters (asphalt and voids content) are shown in Table 8. The designed temperatures were as follows: 200 °C for aggregate heating, 165 °C for conventional asphalt, and 180 °C for asphalt rubber heating.

The mixture was compacted by the repeated passage of a vibrating cylinder over the slabs until the apparent density was achieved, as defined in the design. Then, samples (50 mm × 63 mm × 380 mm) were obtained by slab sawing to perform modulus and fatigue tests.

## 3. Results

### 3.1. Dynamic Modulus and Fatigue Resistance

Dynamic modulus was evaluated in laboratory tests using four-point bending tests, according to the ASTM D3497 standard [52], and testing was carried out at three temperatures (15, 20 and 25 °C) and seven frequencies (10, 5, 2, 1, 0.5, 0.2 and 0.1 Hz). The phase angle was also measured.

Fatigue tests followed the AASHTO TP 8 standard [53] in the controlled strain mode and were conducted at 20 °C and 10 Hz. For each mixture, nine samples were tested at three strain levels (200, 400, and 800 *×* 10^−^^6^ μm/μm). The adopted failure criterion was the loading cycle at which stiffness was reduced to 50%, measured by considering the 100th cycle as the initial cycle [19,53]. The fatigue law is described in Equation (1), where N is fatigue cycles, ε is the tensile strain (10^−^^6^) and a and b are experimental coefficients:N = a × (1/ε)^b^(1)

Table 9 shows the dynamic modulus and phase angle results at 20 °C and 10 Hz (typical temperature and frequency for pavement design). From the analysis of the results, it was possible to affirm that the addition of crumb rubber influenced the reduction of the dynamic modulus. This reduction was confirmed by the highest dynamic modulus being recorded for the conventional asphalt mixture (DNIT5070). An increase in the dynamic modulus was also observed with higher stiffness of the base asphalt, i.e., when PEN 50/70 was changed to PEN 30/45. The phase angle was very low for all mixtures, and no significant influence on the design parameters was found.

The dynamic modulus values obtained at different temperatures and frequencies were used to construct the master curve using the Williams–Landel–Ferry temperature–time superposition model [54], which is expressed in Equation (2), where α_T_ is the shift factor (Equation (3)), Tr is the reference temperature, T is the temperature, C_1_ and C_2_ are parameters dependent on the reference temperature (non-dimensional), and f_r_ is the reduced frequency at reference temperature Tr:f_r_ = fx α_T_(2)
Log (α_T_) = [−C_1_ × (T − T_r_)]/[C_2_ + (T − T_r_)](3)

Figure 5 presents the master curves of the dynamic modulus for the reference temperature of 20 °C. From the master curves, it was possible to observe significant differences between all mixtures at low and highly reduced frequencies. Four asphalt rubber mixtures were tested at only one temperature, for which the master curves are not presented. Comparative to the reference, asphalt rubber mixtures presented a lower dynamic modulus at high frequencies and low temperatures, while the modulus was elevated at low frequencies and high temperatures.

Fatigue tests were performed to determine the fatigue laws, with the results shown in Figure 6. Considerable differences between all mixtures were observed in terms of the obtained fatigue laws. At low strain levels, around 200 *×* 10^−^^6^, the asphalt rubber’s fatigue life varied up to 100-fold in comparison with the reference.

Fatigue life curves were drawn to define the fatigue laws (a linear relationship between the loading cycles and the tensile strain, plotted in the log) according to Equation (1). The parameters of these fatigue laws are presented in Table 10. The number of cycles for the 100 *×* 10^−^^6^ mm/mm strain level (N_100_) was also obtained for use as a comparison parameter. The comparison using the N_100_ parameter (Table 10) showed that the fatigue life of the asphalt rubber mixtures, for 100 *×* 10^−^^6^ mm/mm, ranged from 1.66 × 10^7^ to 2.60 *×* 10^9^ (155 times), meaning that the design’s parameters affected and were responsible for the fatigue performance of these mixtures.

Figure 7, Figure 8 and Figure 9 present how the fatigue resistance (N_100_) is influenced by the design parameters of the asphalt rubber mixtures, such as base asphalt, crumb rubber type, gradation, volumetric parameters, and the production system. In these figures, horizontal arrows indicate the comparisons for identifying the influence of design parameters on fatigue resistance.

According to Figure 7a, the dense (AI) and gap (Caltrans and ADOT) gradations presented similar fatigue resistances when other parameters were fixed (see CA5017CB vs. AA5017CB; CC3017CB vs. IC3017CB; AA5020TB vs. CA5020TB), but there were also some differences (see CA3017CB vs. IA3017CB; AA5017CB vs. IA5017CB). This means that the aggregate gradation is not a parameter for classification of fatigue resistance.

On the other hand, despite an increase in asphalt content improving the fatigue resistance, Figure 7b shows that other factors significantly influenced fatigue resistance, with the highest fatigue resistance obtained for 7% asphalt content while the asphalt mixture with 9% presented one of the lowest levels of resistance.

Figure 8a shows how the voids’ content (air voids) influenced fatigue resistance; the results were not conclusive for distinguishing the mixtures. However, it seems that the asphalt mixtures with 6% air-void content presented less fatigue resistance than those with 5%. Regarding the type of crumb rubber (Figure 8b), the mixtures in this study produced with ambient grinding presented better fatigue performance in general. This fact can be observed by comparing CC5017CB with CA5017CB, and IC3017CB with IA3017CB.

In terms of the base asphalt, it is not clear how it influenced the asphalt rubber mixtures because, as indicated in Figure 9a (see CA3017CB vs. CA5017CB; CC5017CB vs. CC3017CB; IA5017CB vs. IA3017CB), different influences were observed. In these two cases, a stiffer asphalt base contributed to the enhancement of fatigue resistance in the asphalt rubber mixtures. The asphalt rubber production system influenced fatigue performances in an positive manner. Asphalt mixtures from the terminal blend system presented better fatigue resistance (Figure 9b), and this was also observed for gap mixtures (ADOT and Caltrans gradations) and for the dense mixture (AI gradation).

The analysis allowed conclusion that the design parameters influenced the prediction of fatigue resistance, but that this influence also depended on other parameters. Thus, it is impossible to predict the behavior of asphalt rubber mixtures based only on the design parameters. However, these mixtures presented superior mechanical resistance compared with the reference material.

### 3.2. Pavement Design

The previous section has shown that the studied asphalt rubber mixtures have high fatigue resistance, representing clear benefits for pavement performances. However, as asphalt rubber mixtures in a pavement structure have a lower dynamic modulus (relative to the reference), the tensile strain at the bottom of the surface layer tends to be more elevated, reducing the pavement life. In this respect, it is essential to conduct a pavement analysis to predict their behavior.

Thus, to evaluate the asphalt rubber mixtures’ performances as surface layer, pavement structures were designed to estimate the necessary thickness for variations in traffic level. The hypothetical pavement structure had the following layers and respective modulus and thickness: surface (according to the mixture), granular base (200 MPa; 20 cm), and subgrade (100 MPa).

The tensile strain (at the bottom of the asphalt layer) was calculated by JPav software. Equation (4), developed by Pais et al. [55], represents the relationship between the asphalt layer’s thickness and its life. In Equation (4), h denotes the asphalt layer’s thickness (m), N denotes the number of cycles to failure, calculated with the fatigue laws for each material (Table 10), and a, b, and c are coefficients determined with the least squares method, which minimizes the sum of the squares of the errors:log(h) = a + bxlog(N)^2^ + c/log(N)(4)

As a result, the best approximations for the coefficients and the R^2^ values are presented in Table 11. Figure 10 illustrates the pavement thickness as a function of the traffic level for the studied mixtures. As expected, the asphalt rubber mixtures presented superior performance than the reference (DNIT 50/70), except for CA3017CB and CC5017CB mixtures for traffic greater than 1 × 10^7^ ESALs. For traffic less than 1 × 10^7^ ESALs, these two mixtures were found to require almost the same thickness as the reference mixture.

For 10 cm of the reference mixture, the pavement supports 1.7 *×* 10^6^ ESALs. CA3017CB and CC5017CB presented similar performances, 2.17 *×* 10^6^ and 1.97 *×* 10^6^, respectively. The performance results for the other asphalt rubber mixtures were between 8.27 *×* 10^6^ (AA5017CB) and 1.05 *×* 10^8^ (IA5015TB). The three mixtures produced with the terminal blend asphalt presented better performance at about 10^8^ ESALs.

### 3.3. Sustainable Pavements

The sustainability analysis of a product or process includes three main aspects: environmental, social, and economical. Environmental sustainability is usually evaluated via a life-cycle cost analysis based on recycling and reuse strategies and costings for materials and processes. Social factors are analyzed by considering the costs to users and any harmful health effects.Economic factors are considered by assessing the pavement’s costs relative to construction and maintenance operations.

Mechanical performance evaluation of the asphalt rubber mixtures was achieved as the study’s primary objective. Moreover, this paper also showed the effective contribution of this mixture to sustainability in pavement design.

For the studied mixtures, the thickness of the surface layer was calculated, and the reduction in the pavement structure’s total thickness can be considered a benefit with respect to sustainability. The average thickness of the surface layer was considered for three traffic levels (10^6^, 10^7^, and 10^8^ ESALs), and reductions from 50% up to 65% were observed (Figure 11) compared with the reference mixture. This alternative represents a substantial decrease in the cost of pavement construction and raw materials required to provide the same pavement life.

Reduction in the surface layer’s thickness after application of asphalt rubber mixture has previously been investigated and proposed in other research studies [15,56,57].

It is important to emphasize that the mixtures were produced in the laboratory, and the performance results must be validated in the field. Considering environmental issues, one of the essential requirements for constructing efficient roads is executing projects that integrate sustainability. Emissions measurements were not performed in this study, but some authors have already reported the adverse and harmful effects of asphalt rubber emissions [26,30,33,34].

On the other hand, the asphalt rubber mixtures applied on road pavements utilize vast amounts of disposed scrap tires in an environmentally appropriate manner, contributing to reducing their disposal in landfills or illegal areas.

Considering that 1250 scrap tires are used per lane kilometer in a five-centimeter-thick asphalt rubber mixture surface (10% crumb rubber content) [17], mixtures with 15% and 20% (crumb rubber content) similarly represent 1875 and 2500 disposed scrap tires, respectively.

## 4. Conclusions and Recommendations

This study evaluated the effective contribution of asphalt rubber in enhancing the fatigue performance of asphalt mixtures. Crumb rubber obtained from scrap tires was incorporated into the asphalt base to produce asphalt rubber, providing an alternative to disposing of this waste material in the environment.

All of the asphalt mixtures produced were tested at the laboratory to obtain the fatigue laws and the dynamic modulus (stiffness). Then, the results were compared with a reference mixture (unmodified asphalt). Regarding mechanical performance, the following findings could be drawn. The reference mixture presented the highest dynamic modulus compared with the asphalt rubber. The lower modulus of the asphalt rubber mixtures can be attributed to the flexibility of the rubber that was incorporated The phase angle was similar in all mixtures. All asphalt rubber mixtures presented higher fatigue resistance than the reference. The dense-graded asphalt rubber mixture produced with the terminal blend system with 15% crumb rubber content obtained better fatigue performance. The gap-graded mixtures with 20% crumb rubber (terminal blend system) also performed well. The mixture with the worst fatigue life was the reference mixture.

Another contribution of this study was the mechanistic analysis performed from the viewpoint of pavement design. The pavement underlayer’s (base and subgrade) characteristics were maintained (material, modulus and thickness), and the surface layers (asphalt mixtures) were changed according to the obtained results at the laboratory (dynamic modulus and fatigue laws). As a result, for the same operational lifetime (ESALs—equivalent single axle loads), asphalt rubber mixtures as a surface layer, had a pavement thickness reduction of 50% to 65% (surface layer) compared with the reference. The thickness reduction for the same expected life represents the mixture’s ability to save energy and raw material, and provides a potential contribution to the sustainability of road pavements.

The main contribution of this study is related to sustainability, obtainable by asphalt mixtures being modified as part of scrap tire sub-product development and maintaining adequate fatigue performance. The potential to reduce the surface layer’s thickness could minimize harmful environmental effects.

This study also had limitations and can be complemented based on the following recommendations: (i) the mixtures’ fatigue performance must be validated in the field; (ii) it is essential to evaluate the asphalt’s rheological behavior; (iii) emissions arising during the production of asphalt rubber, as well as from mixtures, must be measured; (iv) other mechanical tests must be conducted, such as permanent deformation and crack propagation testing, to complement the fatigue analysis; and (v) a cost analysis should be conducted.

Finally, the superior performance obtained by asphalt rubber mixtures does not represent the solution to all pavement problems. However, its use as a surface layer can be a viable alternative for prolonging pavement life.

## Figures and Tables

**Figure 1 materials-15-08592-f001:**
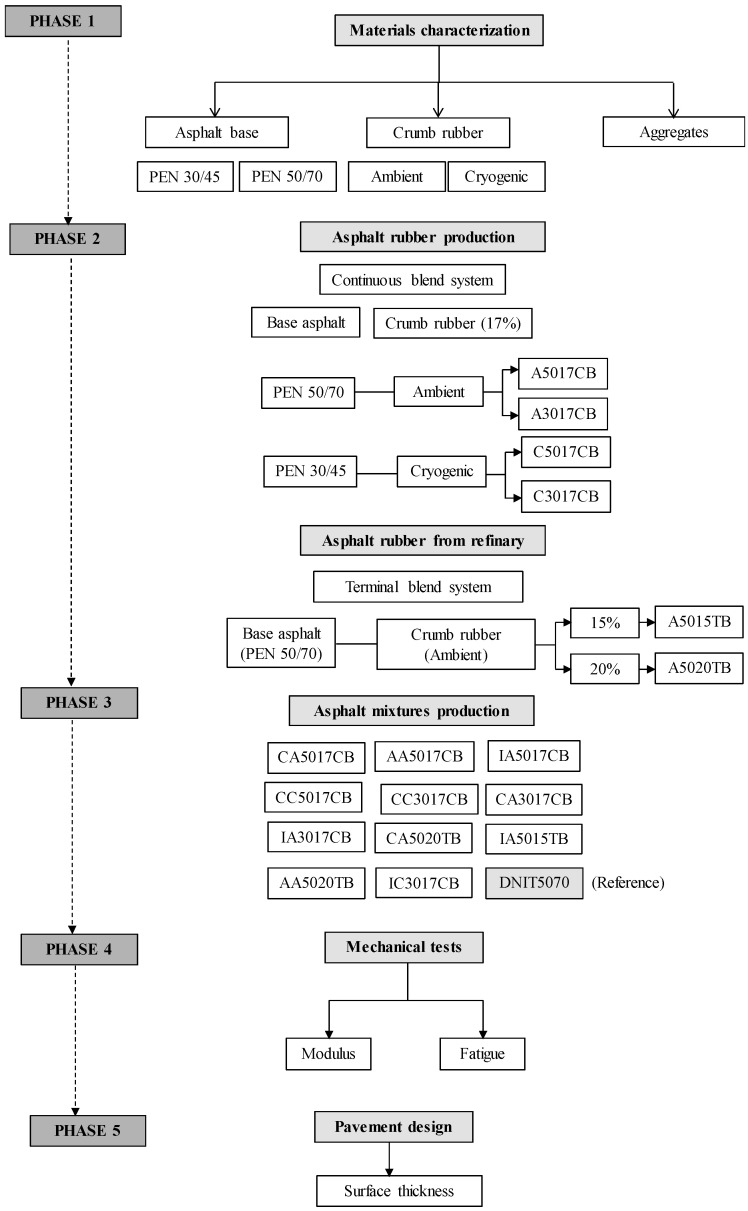
Flowchart of the method.

**Figure 2 materials-15-08592-f002:**
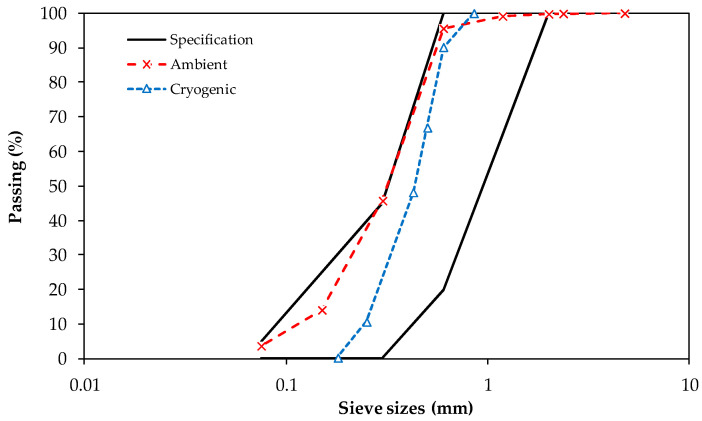
Crumb rubbers gradations.

**Figure 3 materials-15-08592-f003:**
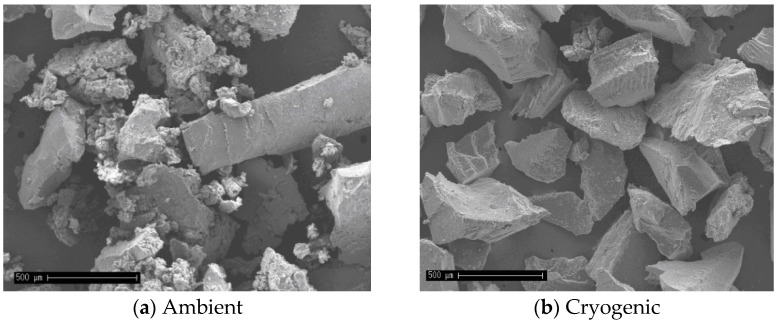
Crumb rubber SEM micrographs for: (**a**) Ambient crumb rubber; (**b**) Cryogenic crumb rubber.

**Figure 4 materials-15-08592-f004:**
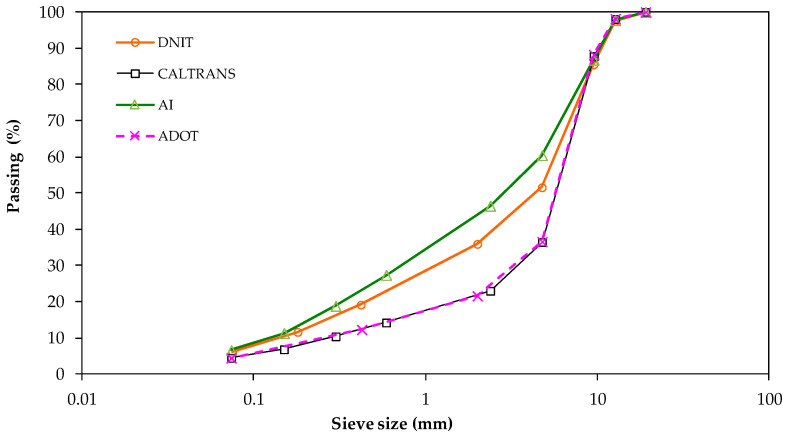
Mixtures’ gradation curves.

**Figure 5 materials-15-08592-f005:**
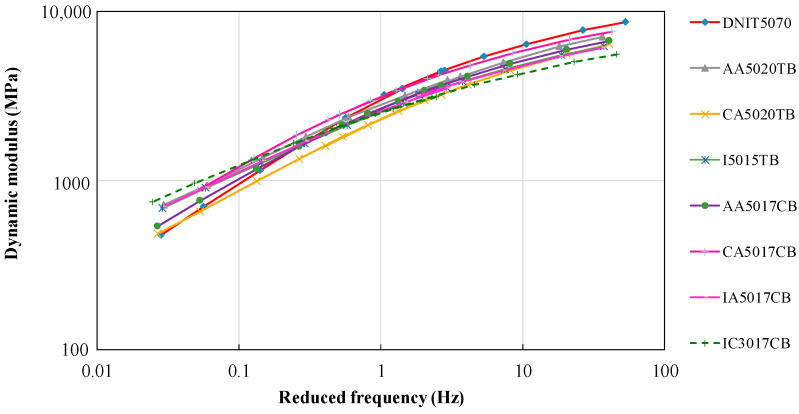
Master curves of dynamic modulus for a reference temperature of 20 °C.

**Figure 6 materials-15-08592-f006:**
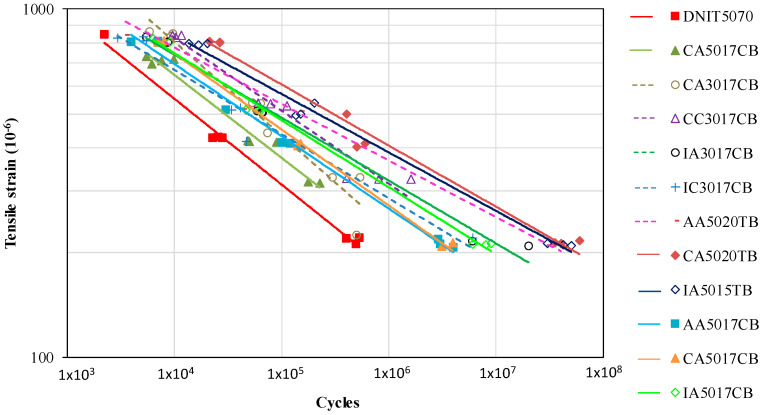
Fatigue curves of the mixtures.

**Figure 7 materials-15-08592-f007:**
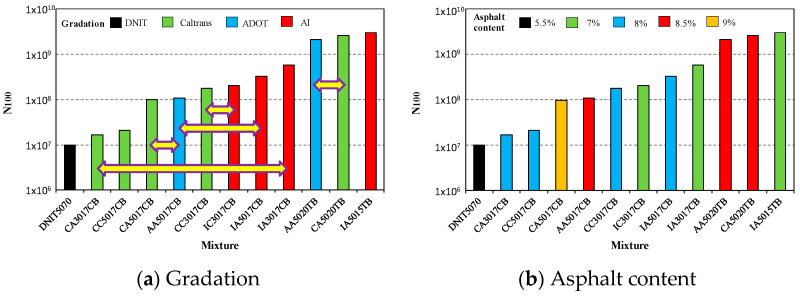
Influence of aggregate gradation (**a**) and asphalt content (**b**) on fatigue resistance.

**Figure 8 materials-15-08592-f008:**
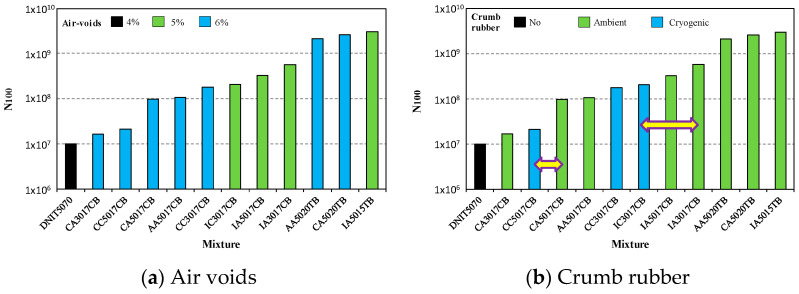
Influence of air voids (**a**) and crumb rubber (**b**) on fatigue resistance.

**Figure 9 materials-15-08592-f009:**
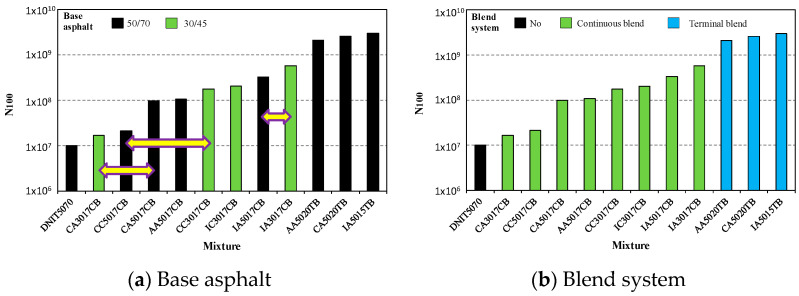
Influence of base asphalt (**a**) and blend system (**b**) on fatigue resistance.

**Figure 10 materials-15-08592-f010:**
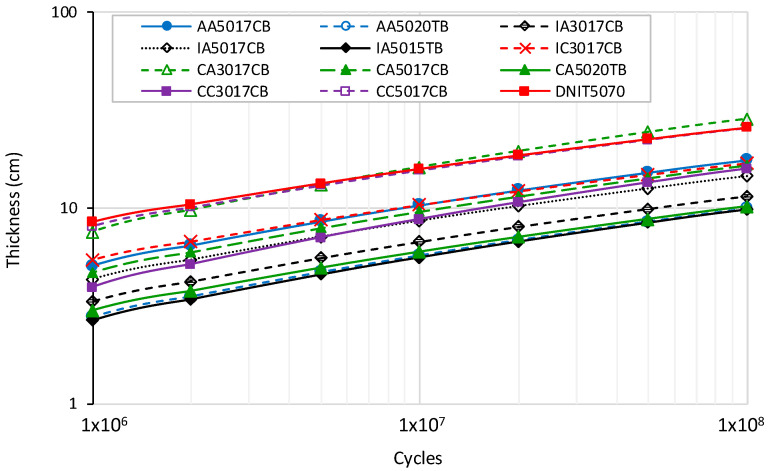
Required thickness of each mixture for pavement design.

**Figure 11 materials-15-08592-f011:**
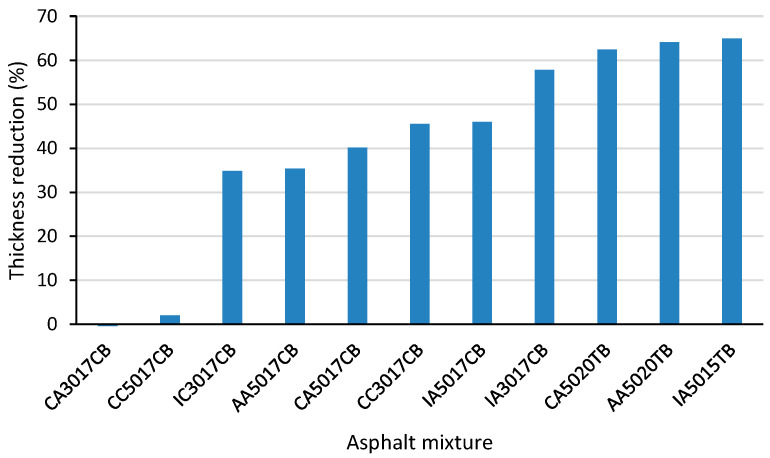
Pavement thickness reduction by using asphalt rubber mixtures.

**Table 1 materials-15-08592-t001:** Asphalt rubber designations.

Designation	Asphalt Base	Crumb Rubber	Rubber Content (%)	System Type
A5015TB	PEN 50/70	Ambient	15	Terminal blend
A5020TB	PEN 50/70	Ambient	20	Terminal blend
A5017CB	PEN 50/70	Ambient	17	Continuous blend
C5017CB	PEN 50/70	Cryogenic	17	Continuous blend
A3017CB	PEN 30/45	Ambient	17	Continuous blend
C3017CB	PEN 30/45	Cryogenic	17	Continuous blend

**Table 2 materials-15-08592-t002:** Asphalt mixture designations.

Asphalt Mixture	Gradation	Asphalt Type
IA5015TB	AI [40]	A5015TB
AA5020TB	ADOT [39]	A5020TB
CA5020TB	Caltrans [38]	A5020TB
IA5017CB	AI [40]	A5017CB
AA5017CB	ADOT [39]	A5017CB
CA5017CB	Caltrans [38]	A5017CB
CC5017CB	Caltrans [38]	C5017CB
CA3017CB	Caltrans [38]	A3017CB
CC3017CB	Caltrans [38]	C3017CB
IA3017CB	AI [40]	A3017CB
IC3017CB	AI [40]	C3017CB
DNIT50/70	DNIT [37]	PEN50/70

**Table 3 materials-15-08592-t003:** Conventional asphalt base characterization.

Property	Standard	Results	
		50/70	30/45
Penetration ^a^ (0.1 mm)	ASTM D5 [41]	51.5 [50–70]	33 [35–50]
Softening point ^b^ (°C)	ASTM D36 [42]	51.5 [50 min.]	52.7 [50–78]
Resilience (%)	ASTM D5329 [43]	0 [n/s] ^c^	9 [n/s] ^c^
Apparent viscosity ^d^ (cP), 135 °C	ASTM D2196 [44]	1580 [274 min.]	175 [n/s] ^c^
RTFOT ^e^ 163 °C, 85 min.	ASTM D2872 [45]		
Change in mass (%)		0.3 [0.5 max.]	0.2 [0.5 max.]
Change in softening point (°C)		4.3 [8.0 max.]	0.5 [n/s] ^c^
Penetration (0.1 mm)		22.3 [n/s] ^c^	27.7 [n/s] ^c^
Retained penetration (%)		43.3 [55 max.]	44.0 [53 max.]

^a^ 100 g, 5 s, 25 °C; ^b^ ring and ball method; ^c^ n/s—not specified; ^d^ Brookfield viscometer, spindle 27, 20 rpm; ^e^ rolling thin film oven test. The asphalt specifications [35,36] are shown in brackets.

**Table 4 materials-15-08592-t004:** Results of coarse aggregate characterization tests.

Property	Standards	Aggregates	Results
Particle shape (flat) (%)	BS 812 [47]	Grade 1	23
		Grade 1	17
Particle shape (elongated) (%)		Grade 2	21
		Grade 2	19
Los Angeles abrasion (%)	ASTM C141 [48]	Grade 1	24
Water absorption (%)	EN 1087-6 [49]	Grade 1	0.88
		Grade 2	1.24
Specific gravity (g/cm^3^)		Grade 1	2.66
		Grade 2	2.65

**Table 5 materials-15-08592-t005:** Results for the fine aggregate characterization tests.

Property	Standards	Results
Methylene Blue test (%)	EN 933-9 [50]	0.02
Sand equivalent Test (%)	EN 933-8 [51]	60
Water absorption (%)	EN 1087-6 [49]	0.41
Specific gravity (g/cm^3^)		2.61

**Table 6 materials-15-08592-t006:** Results of the terminal blend asphalt rubber characterization tests.

Test	Standard	Limits ^f^	A5020TB	A5015TB
Penetration ^a^ (0.1 mm)	[41]	25–75	40	42
Softening point ^b^ (°C)	[42]	54.4 (min)	68.0	67.7
Resilience (%)	[43]	20 (min)	28	33
Apparent viscosity ^d^ (cP), 175 °C	[44]	1500 (min)	2179	1644
RTFOT ^e^ 163 °C, 85 min.	[45]			
Change in mass (%)		0.6 (max)	0.3	0.3
Change in softening point (°C)		n/s ^c^	1.0	2.9
Penetration (0.1 mm)		n/s ^c^	28.8	25.3
Retained penetration (%)		n/s ^c^	72.0	60.2
Apparent viscosity ^d^ (cP), 175 °C	[44]	n/s ^c^	5350	1962
Resilience (%)	[44]	n/s ^c^	39	36

^a^ 100 g, 5 s, 25 °C; ^b^ ring and ball method; ^c^ n/s—not specified; ^d^ Brookfield viscometer, spindle 27, 20 rpm; ^e^ rolling thin film oven test. ^f^ The asphalt specification limits are as stated by ASTM D6114 [16].

**Table 7 materials-15-08592-t007:** Results of the continuous blend asphalt rubber characterization tests.

Test	A5017CB	C5017CB	A3017CB	C3017CB
Penetration ^a^ (0.1 mm)	26.0	36.7	19.7	16.8
Softening point ^b^ (°C)	65.0	55.4	69.9	73.3
Resilience (%)	40	40	52	49
Apparent viscosity ^c^ (cP), 163C	2829	1588	4058	2246
RTFOT ^d^ 175 °C, 85 min.				
Change in mass (%)	0.3	0.3	0.2	0.3
Change in softening point (°C)	8.5	14.7	17.1	11.2
Penetration (0.1 mm)	18.5	21.8	19.5	15.5
Retained penetration (%)	71.1	59.4	99.0	92.2
Apparent viscosity ^d^ (cP), 175 °C	4800	1675	8813	3925
Resilience (%)	46	47	52	56

^a^ 100 g, 5 s, 25 °C; ^b^ Ring and ball method; ^c^ Brookfield viscometer, spindle 27, 20 rpm; ^d^ Rolling Thin Film Oven Test.

**Table 8 materials-15-08592-t008:** Volumetric parameters of asphalt rubber mixtures.

Asphalt Mixture	Air Voids Content (%)	Asphalt Content (%)	Asphalt Mixture	Air Voids Content (%)	Asphalt Content (%)
IA5015TB	5.0	7.0	CC5017CB	6.0	8.0
AA5020TB	6.0	8.5	CA3017CB	6.0	8.0
CA5020TB	6.0	8.5	CC3017CB	6.0	8.0
IA5017CB	5.0	8.0	IA3017CB	5.0	7.0
AA5017CB	6.0	8.5	IC3017CB	5.0	7.0
CA5017CB	6.0	9.0	DNIT50/70	4.0	5.5

**Table 9 materials-15-08592-t009:** Dynamic modulus of the asphalt rubber mixtures (20 °C; 10 Hz).

Asphalt Mixture	Dynamic Modulus (MPa)	Phase Angle (%)	Asphalt Mixture	Dynamic Modulus (MPa)	Phase Angle (%)
IA5015TB	4909	19	CC5017CB	4776	20
AA5020TB	5454	21	CA3017CB	4783	16
CA5020TB	4864	21	CC3017CB	5192	17
IA5017CB	4989	18	IA3017CB	6273	14
AA5017CB	5217	19	IC3017CB	4506	16
CA5017CB	5994	18	DNIT5070	6314	20

**Table 10 materials-15-08592-t010:** Fatigue law parameters (20 °C; 10 Hz).

Asphalt Mixture	a	b	R^2^	N_100_
IA5015TB	2.031 × 10^21^	5.915	0.99	3.00 × 10^9^
AA5020TB	1.475 × 10^21^	5.921	0.96	2.12 × 10^9^
CA5020TB	4.587 × 10^20^	5.623	0.99	2.60 × 10^9^
IA5017CB	6.986 × 10^18^	5.164	0.99	3.28 × 10^8^
AA5017CB	3.925 × 10^17^	4.781	0.99	1.08 × 10^8^
CA5017CB	1.380 × 10^17^	4.574	0.99	9.81 × 10^7^
CC5017CB	3.147 × 10^15^	4.086	0.97	2.12 × 10^7^
CA3017CB	1.711 × 10^14^	3.507	0.94	1.66 × 10^7^
CC3017CB	2.782 × 10^17^	4.597	0.96	1.78 × 10^8^
IA3017CB	4.852 × 10^19^	5.463	0.99	5.75 × 10^8^
IC3017CB	5.269 × 10^18^	5.205	0.96	2.05 × 10^8^
DNIT50/70	1.185 × 10^15^	4.037	0.99	9.99 × 10^6^

**Table 11 materials-15-08592-t011:** Coefficients for Equation (4).

Asphalt Mixture	a	b	c	R^2^
IA5015TB	2.618	5.17 × 10^−4^	−13.266	0.999
AA5020TB	2.573	5.82 × 10^−4^	−12.906	1.000
CA5020TB	2.396	1.40 × 10^−3^	−11.823	0.999
IA5017CB	2.628	7.47 × 10^−4^	−12.107	0.999
AA5017CB	2.884	−1.58 × 10^−4^	−13.032	0.999
CA5017CB	2.758	6.38 × 10^−4^	−12.661	1.000
CC5017CB	2.815	6.83 × 10^−4^	−11.59	0.999
CA3017CB	3.246	−4.29 × 10^−4^	−14.12	0.999
CC3017CB	3.111	−6.94 × 10^−4^	−14.943	0.999
IA3017CB	2.552	8.08 × 10^−4^	−12.348	1.000
IC3017CB	2.596	7.51 × 10^−4^	−11.323	0.999
DNIT50/70	2.665	1.29 × 10^−3^	−10.701	1.000

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
