# Peer review of "Contribution of Asphalt Rubber Mixtures to Sustainable Pavements by Reducing Pavement Thickness"

_materials, 2022, doi:10.3390/ma15238592_

Round 1

Author Response

Dear reviewer,
Our responses are in the attached file.

Reviewer 2 Report

The topic of the paper is interesting and needs to be revised. However, some comments need to be addressed:

1)         The title of this paper is not suitable and General (Sustainable asphalt pavements using asphalt rubber mixtures). I suggest the authors to change it.

2)         In the abstract line 22 to 24 ?  the authors investagted by this mixture which is can reduce the thickness to the half any benchmark for this investigation ?

3)         There is a need to be adequately revised for an abstract section? The abstract is very long, and I suggest rearranging it based on journal requirements. Furthermore, the abstract includes a brief introduction, the aim of study, , problem statement, methodology, and results, making it easy to read for the reader.

4)         Did authors state that the wet process for crumb rubber what about the dry prosess . Please give direct evidence .

5)         The literature review is so crowded with references, and I would expect much more on the technical issues that reflect the CR as mixture. For example, two valuable references. 

  • https://doi.org/10.1007/s42464-020-00050-y

6)         Authors need to give credit for the feasibility of CR application in solving diverse asphalt pavement problems.

7) What do the authors mean by "The reference mixture was produced using 50/70 pen asphalt [55]"?

8) Since the CR-modified asphalt includes heavy metals such as (PAHs) and Zn,, is there any need to restrict any due to the high concentration of toxic chemicals that can affect our health and the environment.

9) In my opinion, the research articles still need to be enhanced since there are many researchers interested to the CR-mixture and the topic is hot for scientific credibility. However, it can be assessed as good, containing certain shortcomings that need to be eliminated. In terms of article quality, I am able to process a repeated review within 3 days in the case of the incorporation of comments or relevant justification for their non-incorporation.

10) When comparing laboratory studies with quality control, which is much more appropriate than at the field site, many researchers confirm that the quality control issue for CR mixtures is still raised. 

Author Response

(The authors gave the same response as above.)

Reviewer 3 Report

The manuscript deals with the rubber asphalt mixture performance. The research is not innovative but it is useful.  The structure of the manuscript is clear. The results are well explained. The manuscript needs proofreading. The method of the manuscript must be improved (it is not clear). The submitted manuscript is lacking on many fronts. Substantial work is required before it can be considered for publication in the Journal of Materials. Some of the important points are highlighted below:

1. I suggest the authors to modify the topic of the manuscript.

2. The abstract should state the purpose of the research, the principal results, and the major conclusions briefly. An abstract is often presented separately from the article, so it must be able to stand alone. The abstract structure should be as follows: (introduction, problem of statement, materials, methods, results, and recommendations). Please revise your abstract.

3. I suggest the authors to conduct a more in-depth review and summarize in section 1.

4. I suggest the authors to correct the flowchart phase 2.

5. I suggest the authors to clarify the main idea of the manuscript since it contains many variables such as different rubber production, different rubber percentages, different mixing method, different aggregate gradation and different types of asphalt binder.

6. Lines 167, I would suggest the authors to cite the mixing method.

7. Line 279, I suggest the authors to check the unit.

8. I suggest the authors to delete section 4.2. Since there is no result in the manuscript related to cost or greenhouse gas emissions.

9. I would suggest the authors to correct the citation style.

10. I suggest the authors to replace old references with new references.

11. The points presented in the conclusion section are not up to the mark. The authors are advised to revise it completely and try to present information, which is a summary of the important aspects discussed in the preceding sections. It certainly lacks in its current form.

- Change the name of section 5 to “Conclusion and Recommendations”.

- Add Recommendations. 

- Make the conclusion in points.

- Add more critical points.

Author Response

(The authors gave the same response as above.)

Round 2

Reviewer 2 Report

The authors have elaborately addressed all the issues and concerns raised by the reviewers. 

Reviewer 3 Report

I have 2 comments only:

I suggest the authors to make the conclusion in points.

I suggest the authors to check the figures and format of the journal.